# Chain Reversion for Detecting Associations in Interacting Variables—St. Nicolas House Analysis

**DOI:** 10.3390/ijerph18041741

**Published:** 2021-02-11

**Authors:** Michael Hermanussen, Christian Aßmann, Detlef Groth

**Affiliations:** 1University of Kiel, Aschauhof, 24340 Eckernförde-Altenhof, Germany; 2Chair of Survey Statistics and Data Analysis, Otto-Friedrich-Universität Bamberg, 96045 Bamberg, Germany; christian.assmann@uni-bamberg.de; 3Leibniz Institute for Educational Trajectories, 96047 Bamberg, Germany; 4Institute of Biochemistry and Biology, University of Potsdam, 14476 Potsdam-Golm, Germany; dgroth@uni-potsdam.de

**Keywords:** St. Nicolas House Analysis, association chains, bivariate correlation coefficients, network graphs, data matrices

## Abstract

(1) Background: We present a new statistical approach labeled as “St. Nicolas House Analysis” (SNHA) for detecting and visualizing extensive interactions among variables. (2) Method: We rank absolute bivariate correlation coefficients in descending order according to magnitude and create hierarchic “association chains” defined by sequences where reversing start and end point does not alter the ordering of elements. Association chains are used to characterize dependence structures of interacting variables by a graph. (3) Results: SNHA depicts association chains in highly, but also in weakly correlated data, and is robust towards spurious accidental associations. Overlapping association chains can be visualized as network graphs. Between independent variables significantly fewer associations are detected compared to standard correlation or linear model-based approaches. (4) Conclusion: We propose reversible association chains as a principle to detect dependencies among variables. The proposed method can be conceptualized as a non-parametric statistical method. It is especially suited for secondary data analysis as only aggregate information such as correlations matrices are required. The analysis provides an initial approach for clarifying potential associations that may be subject to subsequent hypothesis testing.

## 1. Introduction

Scientists often wish to learn more about the association between variables they have obtained in their studies. Anthropologists, epidemiologists, public health specialists, and others may undertake field studies and then apply standard statistical procedures to their data and analyze associations such as weight-for-height to find out about the prevalence of obesity, or they are interested in the correlation between body height and household income, or between skinfold thickness and physical activity, or other variables. The true underlying relationship between these variables usually remains unknown and there is no apparent way to investigate causality within such associations, such as: did household income influence height, or do the taller people enjoy more household income, or did more physical activity lead to a reduction in skinfold thickness, or are less obese persons fitter and therefore, undertake more physical activity [1].

Scientists use various approaches to uncover associations between variables. Apart from looking at isolated pairwise associations between variables using their correlations, linear mixed models may be a way for analyzing data that are non-independent, multilevel/hierarchical, longitudinal, or correlated. There are more sophisticated approaches like Ridge [2] and Lasso regression [3], but these are not widely used by researchers because, perhaps, of the need for complex parameter selection and the required data normalization. Other investigators prefer structural equation modeling including path analysis that tests whether measures of a construct are consistent with a researcher’s understanding of the nature of that construct, or to test whether the data fit a hypothesized measurement model. These tools have in common that they are based on the concept of dependent and independent variables. Principal component, factor or cluster analysis are often used for visualizing dependencies among the variables of interest or grouping them. All these approaches usually fail to answer the question: which is the input, and which is the outcome variable. In addition, all of these methods utilize original sets of raw or carefully normalized data.

We propose an alternative approach. Instead of hypothesis-driven approaches to test and, ultimately, confirm particular assertions for sets of variables, we depict networks of associations between extensively correlated variables. This can be done without detailed information about the original set of raw data. This approach creates an initial foundation for making assertions about the nature of causality within the data set, and may then in a later step be part of hypothesis-driven testing. Our method ranks bivariate correlation coefficients in descending order according to magnitude. Doing so creates chains of associations and is independent of any initial concept of underlying data structure.

Let us assume a small correlation matrix with the variables A, B, C, and D. All variables correlate with each other at various magnitudes. We rank these correlations. We start with A. We find that the correlation, r, between A and B (|r_AB_|) is larger than the correlation between A and C (|r_AC_|), and that the correlation between A and C again is larger than the correlation between A and D (|r_AD_|). We rank the three correlation coefficients in a hierarchic order: |r_AB_| > |r_AC_| > |r_AD_|, and get the chain A-B-C-D. We now invert this sequence and start with endpoint D. We get D-C-B-A. Let us assume that also the sequence of correlation coefficients of the inverted chain |r_DC_|, |r_DB_|, |r_DA_| forms a hierarchic order of magnitude with |r_DC_| > |r_DB_| > |r_DA_|. Sequences of correlation coefficients that are characterized by descending order when starting from either end, are named “association chains”. Association chains can be translated into network graphs. At first view, the whole procedure sounds trivial, but it opens the possibility of immediately visualizing extensive interacting variables in an explorative manner.

As this method successively connects nodes similar to a popular German child game, we named the method St. Nicolas House Analysis (SNHA). German children like to draw St. Nicolas’ house. It consists of five points that need to be connected by one single line (Figure 1) without dropping the pencil. Doing so, the children murmur “this-is-the-house-of-Ni-co-laus”. Except for the top of the roof, each point connects with each other point. Essentially, St. Nicolas house is a network, consisting of “nodes” (circles) and “edges” (lines). Edges direct from one node to the next one. St. Nicolas house is a directed graph that has a topological ordering from earlier to later. We use St. Nicolas house as a model for analyzing variables that are linked by multiple correlations [4].

## 2. Data

We illustrate the suggested approach with three sets of data. The sets were chosen for reasons of simplicity. They were obtained from different backgrounds to make the approach as easily comprehensible as possible.

Sample 1 (33 Olympic decathletes) consists of performance results of 33 Olympic decathletes at the Olympic Games 1988 [5]. The research question is: are performances in the different sport disciplines serially related to each other? Is the performance in 100-m running related to the performing in 110-m hurdles which again is related to the performance in long jump, etc., or is discus throw related to the performance in pole vault which is related to the performance of …, etc. Sample 1 may not be considered representative of any defined population of decathletes. Table 1 illustrates the first six measurements. No information was available on how the measurements were obtained, nor on measurement accuracy. The data were chosen to illustrate chains of associations within a complex network of medium to high correlations.

Sample 2 (UN 1998—United Nations Social Indicators Data 1998) was chosen in view of the topic of this Special Issue—what makes children grow—and consists of 12 social indicators relevant for child health, obtained from 207 nations and published by the United Nations in 1998 [6]. The correlation coefficients tend to be higher than 0.5. The corresponding research question is similar: are the social indicators serially related to each other? The data were chosen to illustrate chains of associations within a complex network of high correlations.

Sample 3 (Birth weight) was also chosen for its relevance to the topic of this Special Issue predicting maternal input variables such as body height, weight, smoking, etc., on the outcome variable birth weight, for investigating risk factors for low birth weight, in 189 children, but in contrast to the samples 1 and 2, is characterized by low correlations, and lacks an a priori assumption regarding input and outcome variables. Sample 3 exemplifies data that are hierarchically structured. The input variables, maternal factors, influence the outcome variable birth weight. The data were collected at Baystate Medical Center, Springfield, Mass during 1986, and may not be considered representative of risk factors for low birth weight. The data were retrieved from the R package MASS [7]. For this sample the research question differs and reads: which of the explanatory (maternal) variables best predict and forecast the future event (birth weight of the child)?

### 2.1. Statistical Analysis

Matrices of bivariate correlations can be transformed into networks of agreement. However, the complexity of such networks may depend on thresholds chosen for the transformation. Linear models, principal component analysis, and cluster analysis are appropriate and commonly used techniques for analyzing multivariate data. Yet, it remains to be elucidated whether networks of bivariate correlations indeed reflect truly existing associations, and if so, which of the commonly used techniques best transform correlation matrices into networks. We exemplify our approach using sample 1:

We transformed the correlation matrix reflecting the agreement between the performance results of 33 Olympic decathletes, into a network by

Depicting all pairwise correlations with a threshold at |r| > 0.3 between the sport performances;Linear models using forward selection for each variable, at two thresholds R^2^ > 0.01 and R^2^ > 0.10;Factor and principal component analysis, and cluster analysis;St. Nicolas House Analysis.

Different statistical techniques result in different networks. We compared the networks by checking for plausibility. Some of the variables are related to fitness of the lower extremities, such as running and jumping, other variables are related to fitness of the upper extremities, such as throwing. Performance results that are related for anatomical reasons, should be closer associated. Yet, plausibility is not proof that either one of the statistical approaches truly provides the correct solution. We therefore added the antagonistic approach for control, and created networks derived from normally distributed, random generated data from 33 virtual Olympic decathletes with zero correlations between the variables in the following way:(1)We scrambled the truly measured results of running, jumping, throwing and randomly re-assigned the scrambled data to each of the 33 Olympic athletes;(2)We created normally distributed random values for running, jumping, throwing with identical mean values and variances for each of the 33 athletes.

Thereafter, we tested the lack of agreement between the zero correlated variables as indicator of falseness, by counting the number of edges within each of the new random based networks.

All calculations were conducted with the programming language R [8] version 3.6.3 which is a free software under the terms of the Free Software Foundation’s GNU General Public License in source code form. Further the following additional freely available R packages, not being part of the standard R installation were used for the analysis, openxlsx [9], asg [10], dgtools [11].

#### 2.1.1. Pairwise Correlations

Often the first step in the analysis of multivariate data is to create a correlation matrix and find out which of the variables are relevantly related to each other. This helps in gauging the degree of agreement between the variables. Figure 2 shows the correlation matrix of the ten disciplines of sample 1.

Correlation matrices can be transformed into networks. Figure 3 displays graphically all correlations of Figure 2, at the thresholds |r| > 0.3 (left) and |r| > 0.5 (right). The disadvantage of such an approach is the arbitrariness when selecting the thresholds. Different thresholds result in different complexity of the network. The essentials of the new method can be summarized as a ranking of bivariate correlation coefficients in descending order according to magnitude (“St. Nicolas House Analysis”, SNHA). SNHA creates hierarchic “association chains” defined by sequences where reversing start and end point does not alter the ordering of elements. In general, correlation matrices, linear mixed model, factor analysis and cluster analysis depict groups of similar variables. Data may share common variance because they may directly influence each other, or there may be common variance due to some unobserved latent source of variation that simultaneously affects the variables within that particular group. All of these methods utilize complete sets of raw data, and essentially focus on dyads of variables that either directly influence each other, or indirectly when unobserved latent variables exist that simultaneously affect clusters of variables thereby suggesting direct effects between variables that in reality, do not exist. In contrast to methods that utilize complete sets of raw data, we are interested in chains of associations that can be derived from correlation matrices. We measure serial effects among variables with extensive interactions. SNHA ranks series of correlation coefficients and thereby creates so-called “association chains”. SNHA does not require original samples of raw data, but only matrices of correlation coefficients. We search for variables that correlate and affect each other serially.

We detect “chains of associations” as follows.

To highlight structural relationships within a set of variables $X_{i},\;i=1,\ldots ,I$, on the basis of the corresponding correlation structure
$$Ρ=\left(\begin{array}{ccc} 1 & \cdots & \rho_{1I}\\ \vdots & \ddots & \vdots \\ \rho_{I1} & \cdots & 1 \end{array}\right)$$
with $\rho_{ij}=|r_{ij}|$, we propose the following procedure to factorize the likelihood function of the variables. Thresholds values in terms of absolute correlation coefficients can be expressed as well as thresholds for the coefficients of determination. For each variable $X_{i}$ taken as a starting point, an association chain can be defined via sorting the variables in terms of descending correlation sequences. The first order neighboring element of variable $X_{i}$ is defined as $c_{i}^{1}=\arg\;\underset{j\neq i}{\max}\rho_{ij}.$ The second order neighboring element of variable $X_{i}$ is defined then as $c_{i}^{2}=\arg\;\underset{c_{i}^{1}\neq j}{\max}\rho_{c_{i}^{1}j}$.

This procedure is repeated until $0.1>\max\;\rho_{c_{i}^{s}j}$ or $\max\;\rho_{c_{i}^{s}j}$ being perceived as insignificant or $c_{i}^{s}\in \left\{i,c_{i}^{1},\ldots ,c_{i}^{s-1}\right\}.$ As a result, we obtain a chain of variables $i,c_{i}^{1},c_{i}^{2},\ldots ,c_{i}^{s}$. To qualify as an association chain, we reverse the order and check whether the same sequence of variables is obtained when starting at variable $c_{i}^{s}$ and receiving i as the corresponding end point. An association chain is defined by any sequences of correlation coefficients, where reversing the start and end point does not alter the order of elements occurring in that sequences. Note that an ordering of two variables occurring in a sequence starting at variable i predetermines the ordering of these two variables in any other sequence.

The validity of the factorizations of the joint density implied by the association chains.
$$f\left(X_{1},X_{2},\ldots ,X_{I}\right)\approx f\left(X_{c_{i}^{s}}|X_{c_{i}^{s-1}},X_{c_{i}^{s-2}},\ldots ,X_{i}\right)f\left(X_{c_{i}^{s-1}}|X_{c_{i}^{s-2}},\ldots ,X_{i}\right)\ldots f\left(X_{i},X_{c_{i}}\right)$$
is assessed via Likelihood Ratio tests, where $X_{c_{i}}$ denotes the set of variables not included in the sequence of variable i. Thereby, the log likelihood of joined unrestricted distribution is compared to the restricted log likelihood implied by the association chain.

Merging all chains discovered within the network forms the final St. Nicolas House graph (Figure 4). The graph comprises seven association chains detected in sample 1. The chains overlap each other and form the final St. Nicolas House graph. The graph plausibly connects the ten sport disciplines according to upper and lower parts of the body. In order to statistically check the plausibility of each of these chains, we calculated log-likelihoods for all chains.

#### 2.1.2. Log-Likelihood Analysis

Seven association chains were detected in the performance results of 33 Olympic decathletes (Table 2). We ask, do these seven chains truly summarize the dependencies among the variables? Is the sequence of performances of chain #1 connecting high jump, 110-m hurdles, and pole vault, a likely result? We performed a likelihood based analysis, and summarized the evidence for the seven association chains detected in 33 decathletes. Noting that the chains correspond to a restricted dependence structure captured in a sequence of conditional distribution as indicated in the boxed text 2, we oppose the restricted likelihood with the unrestricted likelihood, and we allow for all mutual dependencies between the variables entailed within this sequence. The likelihood evidence of restricted dependencies (arising from the chains) and the unrestricted dependence structure can be compared by means of a likelihood ratio test.

Table 2 provides the corresponding results obtained from the association chains detected as shown in Figure 4. We assume either joint normality in case of the unrestricted dependence or conditional normality in case of the restricted dependence as implied by the association chains. This implies that the likelihood ratio test statistic follows an asymptotic ${Χ}^{2}$ distribution.

The ${Χ}^{2}$ test statistics arise here from the log-likelihood differences. In all of these applications, the aim is to test whether some vector parameter is zero vs. the alternative that it is non-zero and the ${Χ}^{2}$ statistic is related to the squared size of the observed effect. The required *p*-value is the right tail probability for the ${Χ}^{2}$ value. The number of restrictions provides the corresponding degrees of freedom (df), where *p*-values below 0.05 indicate significant differences between the restricted and the unrestricted dependencies among the variables entailed within the considered chain. Small *p*-values indicate that the chain is not sufficiently able to capture the dependencies between the variables entailed within the chain. This was not the case. The dependencies within the corresponding sets of variables were found to be sufficiently captured within the chains as all chains detected have *p*-values larger than 0.10 (Table 2).

#### 2.1.3. Model Variance

The correlation between the sport disciplines (Figure 2, sample 1) varies. The edges of the network differ in strength. The ten variables differ in respect to their integration into the network structure. Whereas some disciplines are highly correlated such as disc and shot with r = 0.79, and 100 and 110 with r = 0.61; others almost completely lack any association. In order to estimate the overall predictive power of the network, we created a linear model and calculated a global average R^2^ value.

SNHA algorithm creates association chains by ranking absolute correlation coefficients. Ranking correlation coefficients implies deciding up to which threshold a correlation coefficient may be used for the algorithm. We tested to what extent the threshold (alpha *p*-value) affects the structure of the St. Nicolas House graph. We set four alpha values (threshold at *p* = 0.01 (top left), at *p* = 0.05 (top right), at *p* = 0.10 (bottom left) and at *p* = 0.20 (bottom right). Correlation coefficients below the respective thresholds were not ranked. Figure 5 exemplifies that the number of edges remains quite resilient except for very restrictive thresholds, suggesting robustness of the SNHA algorithm towards the choice of different alpha values. At alpha values *p* = 0.01, SNHA explains 0.441 of the total variance of the sport performances, at alpha values *p* = 0.05 to *p* = 0.20 the algorithm invariantly explains 0.473 of the total variance.

## 3. Results

First, we show the network of agreement between the sport disciplines of sample 1 based on simple correlation thresholds with different |r| (Figure 3). We then analyze the same set of data by statistical methods that are frequently used by scientists when analyzing the association between variables they have obtained in their studies. We apply principal component analysis, cluster analysis and linear modeling. We then compare these methods with SNHA.

In a second step, we apply the same frequently used statistical methods and SNHA to sets of random data that are not correlated. In these data sets, we expect no agreement between the variables, and consider this approach an indicator of falseness whenever spurious correlations are detected.

Finally, we depict the correlation matrices and the SNHA graph of these matrices of the samples 2 and 3 in order to illustrate the dependence structures of interacting variables by SNHA graphs highlighting association chains that may later be subject to subsequent hypothesis testing also in samples that are characterized by low correlations.

Figure 3 depicts the network of agreement between the sport disciplines of sample 1 based on correlation thresholds with |r| > 0.3 (left) and |r| > 0.5 (right). The choice of threshold influences the complexity of the resulting network. Variables referring to the lower extremities are colored red; variables referring to the upper extremities are colored blue. Performance results of lower and the upper extremities are clearly separated. Figure 6 illustrates the Principal Component Analysis of the same sample. The cluster dendrogram similarly separates the ten disciplines and, in addition, highlights the separation of long-distance runs and short-distance sprints.

Figure 7 depicts different network graphs derived from sample 1 (Decathlon), sample 2 (UN98), and sample 3 (Birthweight). The networks are created by pairwise correlations at a threshold of |r| > 0.3; by linear models using forward selection for each variable, at two thresholds of R^2^ < 0.01 (lm 0.01) and R^2^ < 0.10 (lm 0.1); and by SNHA. Whereas networks created by pairwise correlations and linear models using a low threshold, appear somewhat messy and less plausible than the network created by SHNA, neither one of the statistical approaches invokes any a priori guarantee of truth. The low correlated birth weight sample failed when processed by pairwise correlations and linear models using high threshold.

Figure 8 depicts representative examples of virtual networks that lack meaningful correlations between the variables. A few spurious edges within these networks are accidental. The number of edges was used as indicator of falseness.

SNHA analysis both using Pearson (snha.p) and Spearman (snha.s) correlation resulted in the smallest numbers of false random edges (Holm adjusted *p* < 0.05, confirmed by Wilcoxon test), and thus the lowest level of agreement (Figure 9). There was no significant difference in the number of edges created by snha.p and snha.s.

Figure 10 depicts the Spearman correlation matrix for pairwise correlations between the 12 social indicators obtained from 207 nations and published by the United Nations in 1998 (sample 2). The interactions are generally stronger than those of sample 1. Figure 11 depicts the Spearman correlation matrix for pairwise correlations between the maternal risk factors and birth weight of 189 children (sample 3). These correlations are generally comparably low, several are insignificant. Figure 12 depicts the SNHA graphs of the correlation matrices of the samples 2 and 3.

The four variables infant mortality, total fertility rate, maternal education and illiteracy are highly correlated and take a central position. The gender inequality with extremely opposite positions of male and female economic activity is evident. Male education (eduM) appears of marginal importance when compared with the more central position of female education.

The correlations between maternal risk factors and birth weight are markedly lower with R^2^ = 0.071. Nevertheless, the SNHA graph appears plausible. Body weight depends on maternal weight, and is negatively related with uterine irritability, smoking of the mother, and premature labor.

## 4. Discussion

Various statistical approaches exist to uncover associations between variables within large and complicated matrices. Some researchers use linear mixed models, others prefer structural equation modeling, path analysis, principal component, and factor or cluster analysis. Estimating the quality of these models is not trivial and demands advanced techniques such as measuring cluster quality, for instance, using a silhouette index [12]. Nevertheless, it often remains uncertain whether associations between multiply correlating variables that are detected, represent true associations between these variables. Most statistical models are hypothesis-driven in that they are based on the concept of dependent and independent variables. Typically, the majority of these tools rely on sets of original raw or normalized values and cannot be applied when only secondary material such as correlation matrices, is available.

Therefore, we propose a different approach that we call “St. Nicolas House Analysis” (SNHA). We present a method that is able to also re-analyze processed data of which the original material has been lost. We focused on correlation matrices or other symmetric matrices with values measuring the strength of pairwise associations. We do not need the original set of raw data, instead we rank bivariate correlation coefficients in descending order of magnitude. The present approach is not only independent of the raw material, but also independent of any initial concept of underlying data structure. There is no need for a priori assumption regarding input and outcome variables. The data may be hierarchically structured, or not. We rank and create sequences of correlation coefficients named “association chains”. These sequences of ranked correlation coefficients are then translated into network graphs. The approach is non-parametric, it is robust towards outliers, and independent of the distribution of the data.

The resilience of SNHA towards thresholds and its apparent propensity against detecting unwanted unassociated variables is a major advantage over linear modeling, principal component, and factor or cluster analysis. This property is partially due to the fact that spurious significance is more likely to occur in single pairwise associations than in chains connecting several associations in one.

The resulting St. Nicolas House graphs can further be utilized as input for variable predictions using for instance linear or ordinal logistic regression, with calculated R^2^ values for individual nodes, respectively, average R^2^ values for all nodes that can serve as a measure of the quality of how well the graph explains the variances. Association chains may even suggest causality in cases when certain elements of a given chain have a temporal order (direction). A particular direction between elements of an association chain can force adjacent elements of that chain to uptake the same direction. SNHA thus immediately visualizes essential associations between extensive interacting variables and hugely complicated matrices. The method was found particularly convenient when dissecting the impact of parental education, various social variables and nutrition on child growth in studies in Indian and Indonesian school children [4,13,14]. The method is able to create initial foundations for later hypotheses that in a second step may then be part of additional hypothesis-driven testing.

In view of combinatorics—consider the 33 decathletes with their 10 × 10 correlation matrix, the 45 coefficients of correlation allowing for 360 combinations of two, 2520 combinations of three, 15,120 combinations of four, etc. serial coefficients of correlation—it is evident that detecting association chains is not a trivial task and may still need improvement. Future studies referring to log likelihood of single association chains appear promising. Within complex networks, alternative association chains of similar significance can exist and can be made visible by more sophisticated techniques, such as the bootstrap method [15]. The bootstrap method allows for estimating levels of statistical significance for a given St. Nicolas House graph.

## 5. Conclusions

We propose a robust non-parametric statistical tool for detection and visualization of dependencies between extensive interacting variables. The suggested approach labeled as SNHA is based on sequences of the variables with invariant ordering in terms of the underlying association measure. SNHA can be performed also in case there are no individual measurements, i.e., raw data are available, but aggregate information in form of sample moments, e.g., sample correlations or covariances, is available. It thus enhances secondary and reanalysis of published data. The analysis provides an initial approach for clarifying potential associations between variables that may be subject to subsequent hypothesis testing. A graphical representation of the revealed association chains is possible via network graphs. The corresponding R-code is available on request from the senior author of this paper, Dr. Detlef Groth, University of Potsdam.

## Figures and Tables

**Figure 1 ijerph-18-01741-f001:**
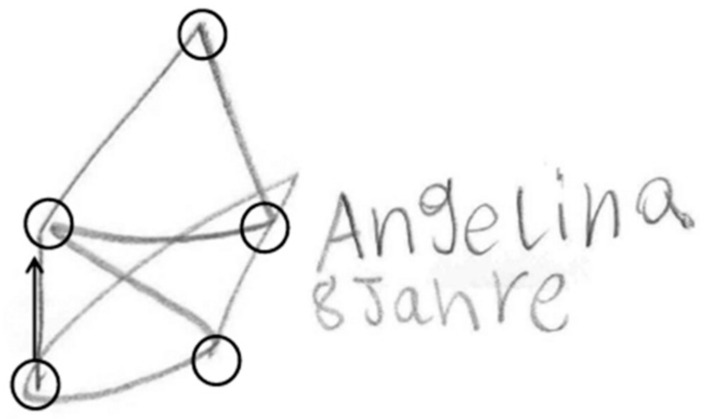
St. Nicolas’ house. Children’s drawing. Nodes are marked by circles. The starting edge is highlighted by arrow.

**Figure 2 ijerph-18-01741-f002:**
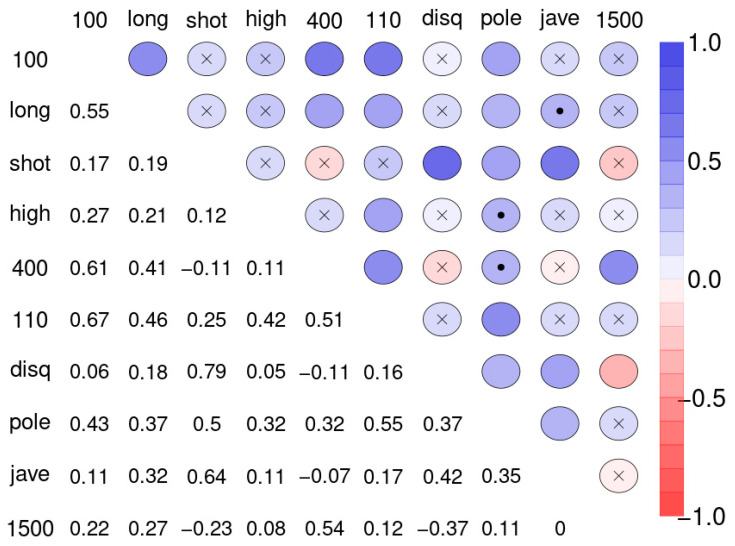
Spearman correlation matrix for pairwise correlations between the disciplines of the decathletes. Upper diagonal: blue color circles indicate positive, red colors indicate negative correlations. Insignificant correlations (*p* > 0.1) are indicated by “x”, weakly significant correlation (*p* > 0.05 and ≤ 0.1) by dot. Lower diagonal: coefficients of correlation (r). Abbreviations used: 100-m sprint (100), long jump (long), shot put (shot), high jump (high), 400-m run (400), 110-m hurdle race (110), discus throw (disq), pole vault (pole), javelin throw (jave), and 1500-m run (1500).

**Figure 3 ijerph-18-01741-f003:**
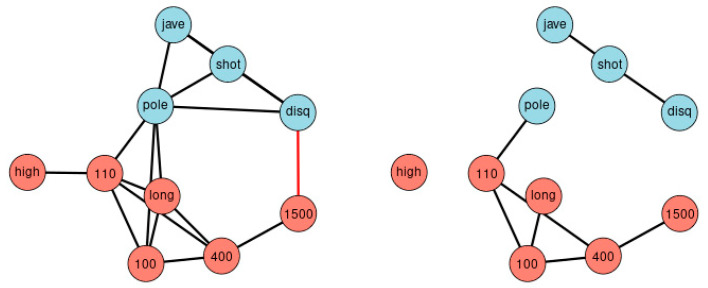
Network of agreement between the disciplines. The network consists of nodes and edges obtained from Figure 2, and depicts all correlation beyond thresholds of |r| > 0.3 (left) and |r| > 0.5 (right). The approach highlights the importance of choosing the appropriate threshold. Blue nodes represent disciplines involving upper parts of the body, red nodes disciplines focus on the lower part of the body. Black edges indicate positive, red edges indicate negative correlations.

**Figure 4 ijerph-18-01741-f004:**
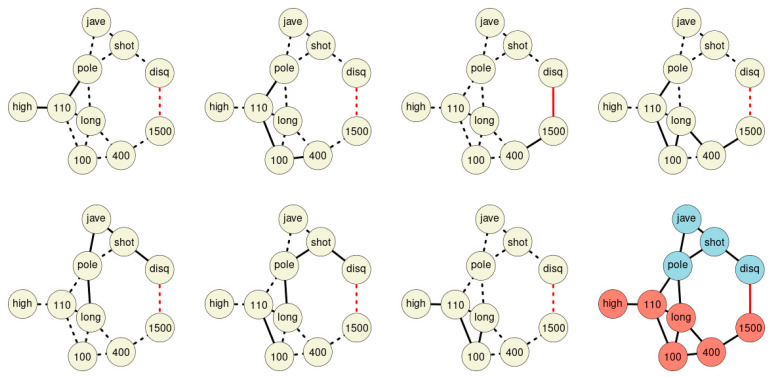
The seven association chains of sample 1. The final St. Nicolas House graph is depicted in the lower right corner. Blue nodes represent disciplines involving upper parts of the body, red nodes disciplines focus on the lower part of the body. Black edges indicate positive, red edges indicate negative correlations.

**Figure 5 ijerph-18-01741-f005:**
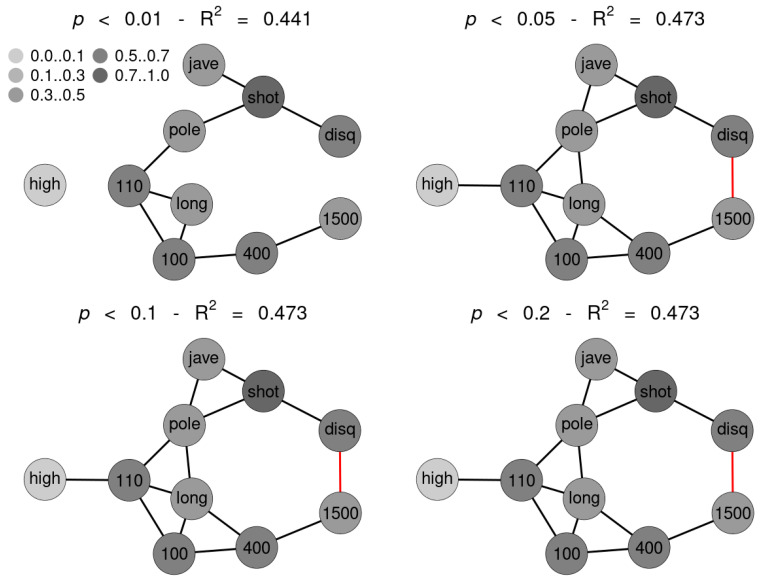
Four St. Nicholas House Analysis (SNHA) graphs (sample 1) for different alpha values (*p*-value threshold at 0.01 (**top left**), at 0.05 (**top right**), at 0.1 (**bottom left**) and at 0.2 (**bottom right**)). Gray shades depict the amount variance that for each variable is explained by SNHA algorithm. R^2^ indicate the total variance of each of the sport disciplines at the given alpha values *p* = 0.01 to *p* = 0.20. similar R^2^ indicates similar validity of the model. The SNHA algorithm is robust towards the threshold chosen. Black edges indicate positive, red edges indicate negative correlations.

**Figure 6 ijerph-18-01741-f006:**
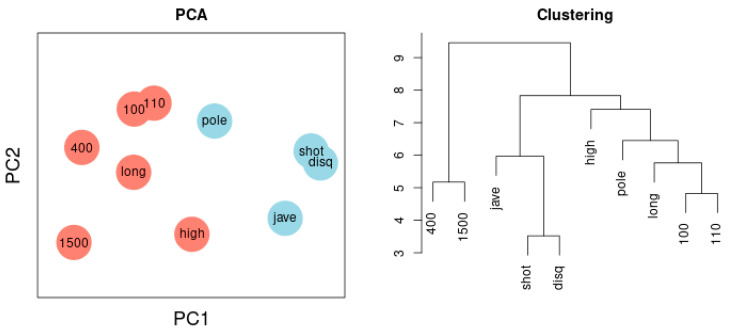
(**Left**) Principal Component Analysis. Two components are separated: lower extremities (red) versus upper extremities (blue). The first component represents 38%, the second 16% of total variance. Throwing discus and 1500-m run are most distant from each other reflecting the negative correlation between the two variables. (**Right**) Cluster dendrogram. Two clusters encompass disciplines reflecting lower extremities and disciplines reflecting upper extremities, a third cluster captures 400-m and 1500-m runs.

**Figure 7 ijerph-18-01741-f007:**
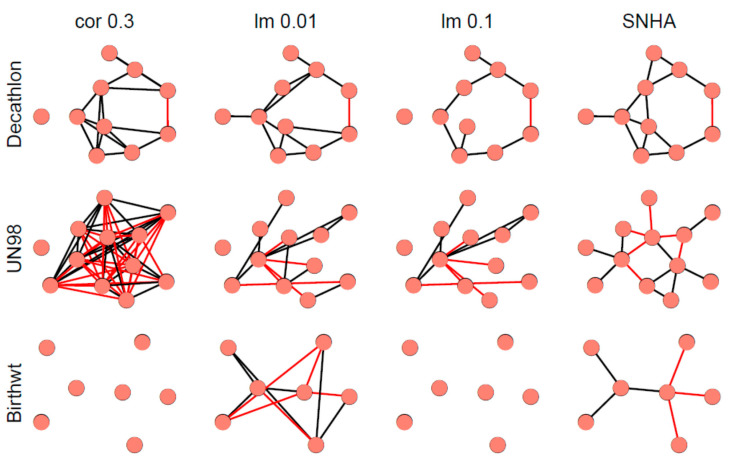
Network graphs derived from sample 1 (Decathlon), sample 2 (UN98), and sample 3 (Birthweight). After excluding non-significant associations (*p* > 0.05) and associations with a predicted R^2^ value of less than 10% and 1%, we created networks by pairwise correlations at a threshold correlation coefficient of |r| > 0.3 (left); by linear models using forward selection for each variable, at two thresholds of R^2^ < 0.01 (lm 0.01) and R^2^ < 0.10 (lm 0.1); and by SNHA. Black edges indicate positive, red edges indicate negative correlations.

**Figure 8 ijerph-18-01741-f008:**
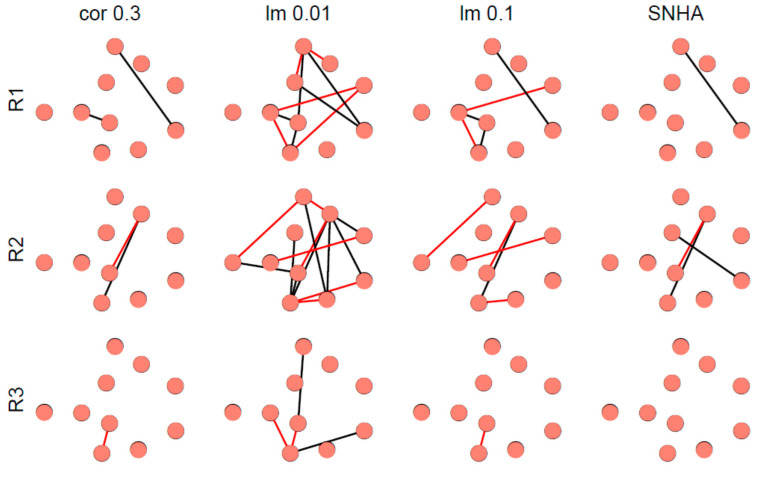
Networks derived from three representative, normally distributed, random generated data sets (R1 to R3) from 33 virtual Olympic decathletes with zero correlations between the variables. Networks were created as described in Figure 7.

**Figure 9 ijerph-18-01741-f009:**
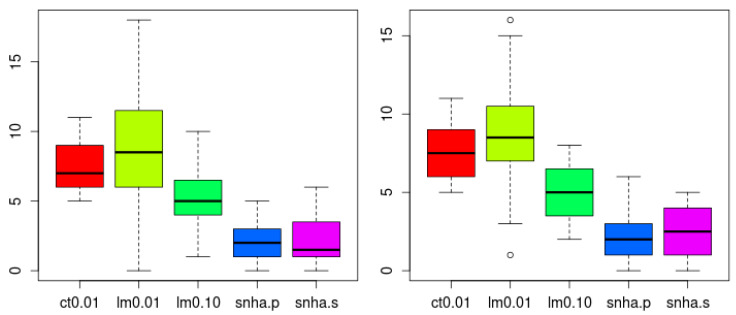
Number of spurious edges within networks created from 20 scrambled (**left**) and 20 normally distributed random data sets (**right**) to indicate falseness. SNHA using Pearson (snha.p) and Spearman (snha.s) correlation resulted in the smallest number of false random edges (less agreement, Holm adjusted *p* < 0.05, confirmed by Wilcoxon test). The number of edges was smaller than predicted by correlation threshold of |r| > 0.1 (R^2^ > 0.01) and linear model building using forward selection with predicted R^2^ value of at least 1% and 10% (lm 0.01, lm 0.10). The number of edges created by snha.p and snha.s (threshold |r| > 0.1), did not differ significantly.

**Figure 10 ijerph-18-01741-f010:**
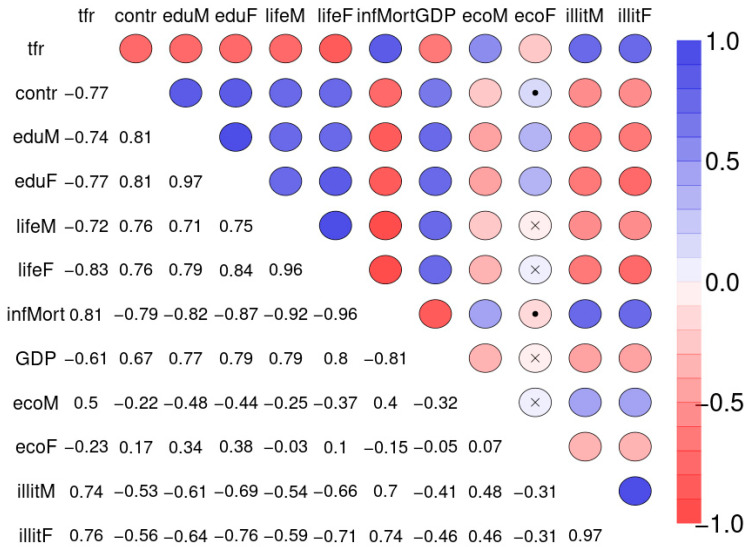
Spearman correlation matrix for pairwise correlations between total fertility rate (number of children per woman, tfr), percentage of married women using any method of contraception (contr), average number of years of father’s (eduM), and mother’s education (eduF), life expectancy at birth for males (lifeM), and females (lifeF), number of infant deaths per 1000 live births (infMort), gross domestic product per person in US dollars (GDP), percentage of men and women who are economically active (ecoM, ecoF), and percentage of males and females age 15 years and older who are illiterate (illitM, illitF).

**Figure 11 ijerph-18-01741-f011:**
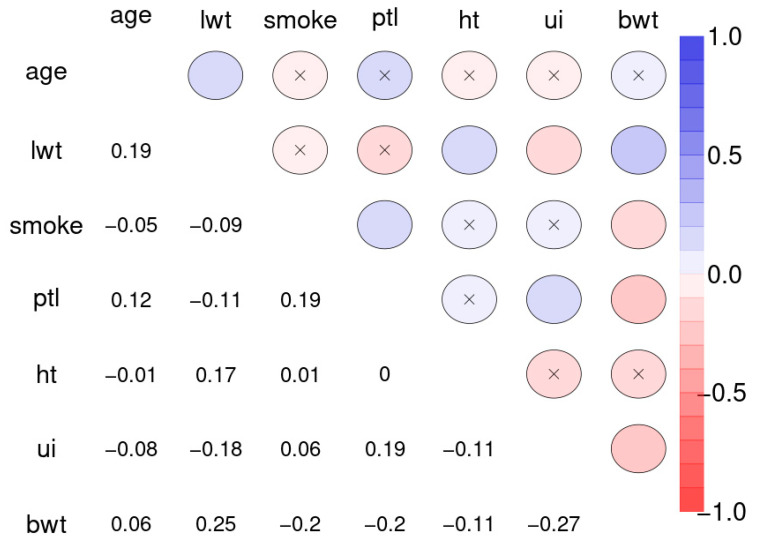
Spearman correlation matrix for pairwise correlations between the maternal risk factors and birth weight of 189 children. Abbreviations used: age: mothers age, lwt: last weight of the mothers at last menstrual period, smoke: status at pregnancy, ptl: premature labor, ht: history of hypertension, ui: uterine irritability, bwt: child birth weight in grams; binary status variables were encoded as 0 for no and 1 for yes.

**Figure 12 ijerph-18-01741-f012:**
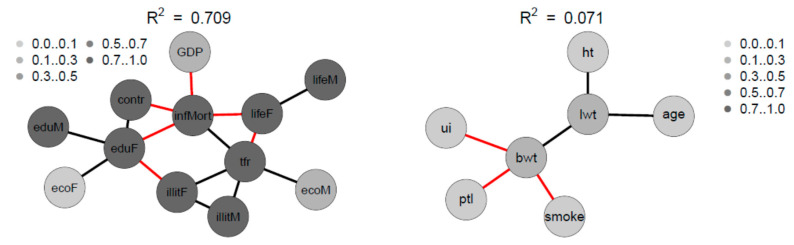
SNHA graph of the correlation matrix of the sample 2 and sample 3. Gray shades depict the amount variance that for each variable is explained by SNHA algorithm. SNHA explains 0.709 of the total variance of sample 2, and 0.071 of the total variance of sample 3.

**Table 1 ijerph-18-01741-t001:** Performance results of the first six out of 33 decathletes at the Olympic Games 1988 [5]. Distances are given in meters, time values for the running disciplines are given as speed (km/h). Abbreviations used: 100-m sprint (100), long jump (long), shot put (shot), high jump (high), 400-m run (400), 110-m hurdle race (110), discus throw (disq), pole vault (pole), javelin throw (jave), and 1500-m run (1500).

Athlete	100	Long	Shot	High	400	110	Disq	Pole	Jave	1500
1	32.00	7.43	15.48	2.27	29.45	26.17	49.28	4.7	61.32	20.08
2	33.12	7.45	14.97	1.97	30.18	27.39	44.36	5.1	61.76	19.78
3	32.20	7.44	14.20	1.97	29.82	26.74	43.66	5.2	64.16	20.52
4	33.90	7.38	15.02	2.03	29.35	26.90	44.80	4.9	64.04	18.94
5	32.67	7.43	12.92	1.97	30.35	27.50	41.20	5.2	57.46	21.04
6	33.24	7.72	13.58	2.12	29.79	27.93	43.06	4.9	52.18	19.70

**Table 2 ijerph-18-01741-t002:** The seven association chains detected in the performance results of 33 Olympic decathletes (sample 1). *p*-values were calculated as the right tail probability of the ${Χ}^{2}$ value.

Chain	Chain Members	Df	Log-Likelihood ${Χ}^{2}$	*p*-Value
#1	high–110–pole	1	0.14	0.71
#2	high–110–100–long	3	2.85	0.41
#3	1500–400–long–100–110–pole	10	15.68	0.11
#4	pole–110–100–400	3	2.48	0.48
#5	100–110–long–pole–shot–disq	10	13.38	0.20
#6	disq–shot–jave–pole–long	6	7.46	0.28
#7	400–1500–disq	1	0.51	0.48

## Data Availability

Decathlete data (sample 1) were obtained from [5], UN 1998—United Nations Social Indicators Data 1998 (sample 2) were obtained from [6], birthweight data (sample 3) were obtained from [7,8]. Packages [10,11] are available on request by senior author.

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
