# Peer review of "Chain Reversion for Detecting Associations in Interacting Variables—St. Nicolas House Analysis"

_ijerph, 2021, doi:10.3390/ijerph18041741_

Round 1

Reviewer 1 Report

This work focuses on a very interesting topic. I really enjoyed reading this work and I think it is a valid and interesting contribution to the approach of statistical tools that make it possible to analyze associations between variables.

There are some aspects that can be improved:

Lines 2-3 -Title - Consider replacing parentheses with a colon or another solution that highlights the proposed algorithm. Hyphenations of words in the title should be avoided

Line 76 - Review the alleged text box referred to in line 66

Line 96 - Figure 1 must be referenced before appearing

Line 111 -  Section 2 “Data” should start with an introductory paragraph

Line 216 - Figure 3 must be referenced before appearing

Line 217 - Consider using |r| instead of r, in order to also include negative correlations

Line 234 - Review the alleged text box referred to in line 222

Line 275 - Review alignment

Line 284 – Review “[end of text box]”

Lines 289-292 - Clarify and format the caption in Figure 4

Line 371 -  Section 3 “Results” should start with an introductory paragraph

Line 383 - It is convenient to complete the legend in Fig 6, referring to the output relative to the PCA presented (biplot). The presentation of the axes would facilitate the interpretation of the output and give indications on the quality of representation of the categories under analysis

Line 408 - Figure 8 must be referenced before appearing

Line 448 - Figure 11 must be referenced before appearing

Line 454 - Figure 12 must be referenced before appearing

Lines 529-537 - A review of the Conclusion is suggested. It doesn't sound right to start with “St. Nicolas House is a graph.”

The presentation of the results for the 3 samples under analysis requires better organization. A very complete analysis of sample 1 is presented while samples 2 and 3 are not even contextualized, receiving less importance. A more careful organization and selection of the information presented in the sections "2. Data" and "3. Results" should be carried out since the function of these two different sections is not clearly understood.

It would be important to carefully format the entire document and standardize the final list of bibliographic references.

Author Response

We are grateful for the careful reviewing, and in the following, comment on the items raised by the reviewers.

Ref 1

Comments and Suggestions for Authors

This work focuses on a very interesting topic. I really enjoyed reading this work and I think it is a valid and interesting contribution to the approach of statistical tools that make it possible to analyze associations between variables.

There are some aspects that can be improved:

Lines 2-3 -Title - Consider replacing parentheses with a colon or another solution that highlights the proposed algorithm. Hyphenations of words in the title should be avoided

Done

Line 76 - Review the alleged text box referred to in line 66

Done

Line 96 - Figure 1 must be referenced before appearing

Done

Line 111 -  Section 2 “Data” should start with an introductory paragraph

Done

Line 216 - Figure 3 must be referenced before appearing

Done

Line 217 - Consider using |r| instead of r, in order to also include negative correlations

Done. We are very sorry about our negligence.

Line 234 - Review the alleged text box referred to in line 222

Done

Line 275 - Review alignment

Done

Line 284 – Review “[end of text box]”

deleted

Lines 289-292 - Clarify and format the caption in Figure 4

Done

Line 371 -  Section 3 “Results” should start with an introductory paragraph

Done

Line 383 - It is convenient to complete the legend in Fig 6, referring to the output relative to the PCA presented (biplot). The presentation of the axes would facilitate the interpretation of the output and give indications on the quality of representation of the categories under analysis.

This is an important point, and we discussed it thoroughly.

We are well aware that adding details to the fig 6 will improve visualization, but it also adds complexity. As the PCA is just an example of how multiply correlating data can be analyzed, but does not add essentials to the understanding of the center message of this paper, we decided to keep the figure as simple as it is. We are afraid to confuse the reader, and rather favor the present presentation. We hope the argument of greater simplicity for the reader may convince the reviewer.

Line 408 - Figure 8 must be referenced before appearing

Done

Line 448 - Figure 11 must be referenced before appearing

Done

Line 454 - Figure 12 must be referenced before appearing

Done

Lines 529-537 - A review of the Conclusion is suggested. It doesn't sound right to start with “St. Nicolas House is a graph.”

Done

The presentation of the results for the 3 samples under analysis requires better organization. A very complete analysis of sample 1 is presented while samples 2 and 3 are not even contextualized, receiving less importance. A more careful organization and selection of the information presented in the sections "2. Data" and "3. Results" should be carried out since the function of these two different sections is not clearly understood.

Done. We rewrote the respective paragraphs and hope that it has improved

It would be important to carefully format the entire document and standardize the final list of bibliographic references.

Done. Sorry for our negligence

Reviewer 2 Report

The paper is clearly written and will be of interest to the broad readership. The proposed SNHA methodology is non-parametric, deploying intuitive graphical models and enabling quick detection of non-linear associations in the data. An additional attractive feature of SNHA is that it is not prone to false discovery (spurious association), which is especially useful when handling categorical and/or network data. The method is well explained and amply illustrated by real-life examples.

There are occasional minor editorial deficiencies, such as missing spaces etc. The authors should also streamline and unify the list of references to follow the adopted in-house style.

Other than that, I am happy to recommend the paper for publication. 

Author Response

Response to the reviewers.

Ref 2

The paper is clearly written and will be of interest to the broad readership. The proposed SNHA methodology is non-parametric, deploying intuitive graphical models and enabling quick detection of non-linear associations in the data. An additional attractive feature of SNHA is that it is not prone to false discovery (spurious association), which is especially useful when handling categorical and/or network data. The method is well explained and amply illustrated by real-life examples.

There are occasional minor editorial deficiencies, such as missing spaces etc. The authors should also streamline and unify the list of references to follow the adopted in-house style.

Done

Thank you very much again

Michael Hermanussen

Christian Aßmann

Detlef Groth

Round 2

Reviewer 1 Report

The authors considered most of the suggestions and corrections, significantly improving the quality of the manuscript.